# Locked-in Intact Functional Networks in Children with Autism Spectrum Disorder: A Case-Control Study

**DOI:** 10.3390/jpm11090854

**Published:** 2021-08-28

**Authors:** Andrew R. Pines, Bethany Sussman, Sarah N. Wyckoff, Patrick J. McCarty, Raymond Bunch, Richard E. Frye, Varina L. Boerwinkle

**Affiliations:** 1Mayo Clinic Alix School of Medicine, 13400 E Shea Blvd, Scottsdale, AZ 85259, USA; pines.andrew@mayo.edu; 2Division of Neurology, Barrow Neurological Institute at Phoenix Children’s Hospital, 1919 E. Thomas Rd, Ambulatory Building, Phoenix, AZ 85016, USA; bsussman@phoenixchildrens.com (B.S.); swyckoff@phoenixchildrens.com (S.N.W.); vboerwinkle@phoenixchildrens.com (V.L.B.); 3Section on Neurodevelopmental Disorders, Division of Neurology, Barrow Neurological Institute at Phoenix Children’s Hospital, 1919 E. Thomas Rd, Ambulatory Building, Phoenix, AZ 85016, USA; pmccarty@phoenixchildrens.com; 4Division of Psychiatry, Barrow Neurological Institute at Phoenix Children’s Hospital, 1919 E. Thomas Rd, Ambulatory Building, Phoenix, AZ 85016, USA; rbunch@phoenixchildrens.com; 5Department of Child Health, University of Arizona College of Medicine–Phoenix, Phoenix, AZ 85004, USA

**Keywords:** autism spectrum disorder, locked-in network syndrome, resting-state functional magnetic resonance imaging, temporal lobe epilepsy

## Abstract

Resting-state functional magnetic resonance imaging (rs-fMRI) has the potential to investigate abnormalities in brain network structure and connectivity on an individual level in neurodevelopmental disorders, such as autism spectrum disorder (ASD), paving the way toward using this technology for a personalized, precision medicine approach to diagnosis and treatment. Using a case-control design, we compared five patients with severe regressive-type ASD to five patients with temporal lobe epilepsy (TLE) to examine the association between brain network characteristics and diagnosis. All children with ASD and TLE demonstrated intact motor, language, and frontoparietal (FP) networks. However, aberrant networks not usually seen in the typical brain were also found. These aberrant networks were located in the motor (40%), language (80%), and FP (100%) regions in children with ASD, while children with TLE only presented with aberrant networks in the motor (40%) and language (20%) regions, in addition to identified seizure onset zones. Fisher’s exact test indicated a significant relationship between aberrant FP networks and diagnosis (*p =* 0.008), with ASD and atypical FP networks co-occurring more frequently than expected by chance. Despite severe cognitive delays, children with regressive-type ASD may demonstrate intact typical cortical network activation despite an inability to use these cognitive facilities. The functions of these intact cognitive networks may not be fully expressed, potentially because aberrant networks interfere with their long-range signaling, thus creating a unique “locked-in network” syndrome.

## 1. Introduction

Autism spectrum disorder (ASD) has a prevalence of 1 in 54 children in the United States and is characterized by deficits in social communication and interaction, and restricted and repetitive behaviors [1]. The neurological basis of ASD remains elusive in this highly heterogenous population. Resting-state functional magnetic resonance imaging (rs-fMRI) is a promising tool that has been used to analyze neural networks in patients with neurodevelopmental and neurological disorders. A review of prior group-level rs-fMRI investigations suggests hypo-, hyper-, and mixed-connectivity patterns among individuals with ASD, with contradictory findings attributed to differences in age, sex, comorbidities, and/or variations in the rs-fMRI scanning and analysis procedures [2]. Other researchers demonstrated that traditional case-control methods that assume homogeneity within clinical populations lead to a loss of subject-specific features of ASD at the group level and disguise the interindividual variation crucial for precision medicine [3].

To investigate these inconsistencies, a large-scale database replication study that characterized and evaluated connectivity patterns was conducted. The study reports evidence of reproducible ASD-associated functional hyper- and hypo- connectivity linked to clinical symptoms [4]. The authors reported that overall global connectivity was preserved in individuals with ASD, and hyperconnectivity patterns observed in the parietal and prefrontal regions were associated with the severity of deficits in communication and adaptive behavior. Furthermore, they suggested that the connectivity findings support the idea that individuals with ASD are unable to engage or disengage specific networks to the same degree as healthy, typically developing controls. If network engagement capacity is a key mechanism of ASD, then individual-based analysis may elucidate the network engagement potential and allow a better understanding of the clinical heterogeneity of ASD [5].

To address these knowledge gaps in ASD, we used a data-driven rs-fMRI whole-brain-network analysis to identify individual network pathology [6,7], similar to our work in epilepsy [8,9,10]. Independent component analysis (ICA) of rs-fMRI allows for the characterization and visualization of individual resting-state networks (RSNs), both typical and atypical [11,12,13,14]. In epilepsy, we utilize this method clinically to both identify the aberrant RSNs associated with seizure foci and to visualize intact cognitive RSNs. This allows the confirmation of intact cognitive networks and provides information regarding the proximity of intact cognitive networks to aberrant networks [15].

Similarly, in individuals with ASD, we may use rs-fMRI clinically to visualize network patterns in relation to the patient’s clinical phenotype. One feature that differentiates individuals with ASD from those with brain injury is that individuals with ASD can have normal or extraordinary skills despite their disability, and some children with ASD can make substantial improvement with therapies, losing many, if not most, of their ASD symptoms. One of the most enigmatic subsets of ASD includes those that undergo neurodevelopmental regression, suddenly losing normal skills and rapidly developing the ASD phenotype. This suggests that the brains of some individuals with ASD have the capacity to support typical cognitive networks, at least at some time in their life, but may be unable to express these cognitive networks. For children with epilepsy, aberrant networks resulting from ongoing subclinical interictal discharges originating from the seizure onset zone (SOZ) can interfere with the function of typical cognitive brain networks, disrupting their ability to function optimally.

Thus, on the basis of the existing literature and our clinical observations, we hypothesized that some children with ASD may have intact cognitive networks identified on the individual level that are not fully realized, similar to children with epilepsy, perhaps because aberrant networks interfere with their function. However, unlike in children with epilepsy, we hypothesized that these aberrant networks are not localized to a SOZ. To investigate this possibility, subject-level ICA of rs-fMRI was used to identify and characterize the RSNs of children with regressive-type ASD and temporal lobe epilepsy (TLE). The unique relationship between patient and network characteristics was examined.

## 2. Materials and Methods

Using a case-control design, this study examined an age- and sex-matched cohort of children severely affected with regressive-type ASD, and a cohort of children with TLE without ASD. A 2017 meta-analysis reported an ASD pooled prevalence of 6.3% in patients with epilepsy, with a 41.9% risk of ASD in patients with focal seizures (TLE) [16]. Thus, a pathological control group was selected to distinguish ASD-specific atypical rs-fMRI biomarkers from known TLE-specific markers given the increased comorbidity of these conditions. For the ASD cohort, children with regressive-type ASD were chosen since these children had documented normal development early in life, suggesting that their brain could previously support normal cognitive networks. ASD was diagnosed using the Diagnostic Statistical Manual of Mental Disorders (5th ed., DSM-5).

The study sample size was limited due to rs-fMRI data availability for this patient population. Five patients with regressive-type ASD who were not making sufficient progress with standard therapy and five patients with TLE undergoing pre-surgical planning for epilepsy surgery underwent clinically indicated rs-fMRI at Phoenix Children’s Hospital (PCH) between November 2018 and December 2020. The patient cohorts were age- and sex-matched. The PCH Institutional Review Board approved the study, and caretakers provided informed consent to authorize secondary analysis of the rs-fMRI data and review of the medical records.

### 2.1. Resting-State MRI

The rs-fMRI images were acquired and analyzed per prior reported standards [15]. Acquisition was from a 3 T MRI scanner (Ingenuity; Philips Medical Systems, Best, The Netherlands) with a 32-channel head coil. Patients received conscious sedation by propofol, per hospital clinical standards. Acquisition of rs-fMRI consisted of two 10-min runs totaling 20 min. Parameters were 2000 millisecond repetition time (TR), 30 millisecond echo time (TE), 80 × 80 matrix size, 80° flip angle, 46 slices, 3.4 mm slice thickness with no gap, 3 × 3 mm in-plane resolution, interleaved acquisition, and 600 total volumes. For anatomical reference, a T1-weighted turbo field-echo whole-brain sequence was obtained with TR 9 milliseconds, TE 4 milliseconds, flip angle 80°, slice thickness 0.9 mm, and an in-plane resolution of 0.9 × 0.9 mm.

### 2.2. Independent Component Analysis Approach

ICA is driven by empirical data rather than a priori information. Briefly, rs-fMRI voxels are grouped together into components according to similarity of blood oxygen level dependent (BOLD) signal oscillation [6,7]. The resulting independent components (ICs) are independently fluctuating clusters of brain activity or sources of noise that require expert review and interpretation [17]. ICA procedures were completed via the Multivariate Exploratory Linear Optimized Decomposition into Independent Components (MELODIC) tool [7]. The following standard preprocessing steps were applied: (1) deletion of the first five volumes to remove T1 saturation effects, (2) high-pass filtering at 100 s, (3) inter-leaved slice time correction, (4) no spatial smoothing, and (5) motion correction with MCLFIRT [18], with non-brain structures removed. Individual functional scans were registered to the patient’s corresponding anatomical scan using linear registration [19], and optimized using boundary-based registration [20]. All participants had <1 mm head-motion displacement in any direction. As ICA was applied in the subject space, no standardized templates or spatial normalization procedures was performed. The total number of detected ICs was determined for each patient from established automated dimensionality estimates using a Bayesian approach, and an ICA threshold (*p* < 0.05) for IC detection was set by the standard local false discovery rate [6].

### 2.3. Component Categorization

IC categorization followed the working paradigm previously published [15], separating patient ICs into four categories—noise, typical RSNs, SOZ, and atypical (aberrant) networks—using criteria modified from established norms [11,12,13,14,15,21]. Noise components arise from respirations, cerebrospinal fluid movement, and tissue–fluid junctions. A component was determined to be noise if: it was not primarily located in grey matter; it varied significantly in coordination with physiological cycles (i.e., respiratory-related frequency range, 0.1–0.5 Hz; cardiac-related, 0.6–1.2 Hz; regular but fast oscillation pattern); or it was in a spatial distribution consistent with a machine-generated artifact [11,12]. Components determined to be neural networks were categorized as either typical RSNs, SOZs, or atypical (aberrant) networks. Typical RSNs were determined by visually comparing spatial features to established RSNs (e.g., motor, language, and frontoparietal), and comparing temporal features of frequency and frequency power spectra with a known low-frequency, regular, slow-oscillating time course, and low-frequency power-spectra features of RSN norms [12,13,14]. SOZs were distinguished by a spatial pattern (more asymmetrically unilateral than expected, alternating localized activation-deactivation patterns of gray matter, and with a tapered tail from the cortex extending toward the ventricles) not conforming with noise or typical RSNs, an irregular time course, or containing a frequency >0.4 Hz [15]. Aberrant networks were distinguished by spatial locations that may overlap with known RSN, but do not conform to the RSN spatial pattern, noise, or SOZ criteria, having a regular sinusoidal oscillation pattern that is overlaid with irregular faster frequency, and having an atypical BOLD oscillation frequency >0.039 Hz [13,14,15]. rs-fMRI data were interpreted by an rs-fMRI specialist (senior author), wherein typical RSNs were operationally defined as those meeting the spatial and temporal criteria above by expert visual inspection, as in prior publications [12,15].

### 2.4. Statistical Analysis

Categorical variables (e.g., typical RSN: present, absent; aberrant (non-SOZ) network: present, absent) were generated for the motor, language, and frontoparietal networks for each patient. Due to the small sample size, a two-tailed Fisher’s exact test was used to examine the significance of the association between the two factors in the contingency tables. Since, for each set of tests, we compared three cortical regions, requiring three statistical tests, the Bonferroni correction for inflated alpha was used to set the significance threshold at *p* <= 0.017 (0.5/3). An independent-samples t-test (two-tailed, *p* < 0.05) was used for the analysis of continuous variables.

## 3. Results

Figure 1 visually summarizes the clinical characteristics and typical and aberrant rs-fMRI ICA-based networks for children with ASD. Figure 2 demonstrates a detailed example of typical RSNs and aberrant networks for a single ASD case (Patient 1). A narrative summary of the clinical and network features is presented below for all patients, followed by a statistical analysis of patient characteristics and resting-state networks.

### 3.1. Patient Summaries of ASD Patients

#### 3.1.1. Patient 1

A 7-year-old (yo) male progressively lost social interactions and eye contact from 9 to 15 months of age. At the time of rs-fMRI, he was diagnosed with ASD, cognitive delay, severe language impairment (LI), and developmental coordination disorder (DCD). Despite these deficits, frontoparietal (FP), language, and motor RSNs were intact. Aberrant networks were found over sensory, FP, and contralateral non-dominant language regions (Figure 2).

#### 3.1.2. Patient 2

A 4 yo female suddenly lost normal speech, social, language, cognitive, and motor abilities at 3.5 yo. At the time of rs-fMRI, she was diagnosed with ASD, borderline intellectual disability (ID), severe LI, and delayed visual-motor skills. Despite these deficits, FP, language, and motor RSNs were normal. Aberrant networks were found over the right FP, bilateral temporal, and opercular regions.

#### 3.1.3. Patient 3

A 16 yo female lost social skills and ceased making eye contact with others at 3 yo. At the time of rs-fMRI, she was diagnosed with ASD, ID, severe LI, and DCD. Despite these deficits, FP, language, and motor RSNs were detected. Aberrant networks were found over the left FP-temporal regions.

#### 3.1.4. Patient 4

A 12 yo male lost language skills (from saying short sentences to saying only single words) and motor skills (ceased walking and performing fine motor skills) at 2.5 yo. He regained some language and motor skills at 5 yo and began to walk again, but he did not regain fine motor skills. At the time of rs-fMRI, he was diagnosed with ASD, severe ID, moderate LI, DCD, and epilepsy. Despite these deficits, FP, motor and bilaterally-dominant language RSNs were intact. Bilateral language dominance is a relatively mild atypical feature found in children with epilepsy and dominate-sided language region network pathology [22]. Aberrant networks were found over the FP and language regions. The network pathology is consistent with children with drug-resistant epilepsy [8,15] which, in comparison to other aberrant networks discussed, has markedly erratic, high-frequency BOLD time courses and generalized spatial distribution of the cingulate, lateral temporal, and bilateral frontal regions.

#### 3.1.5. Patient 5

A 12 yo male lost language (from knowing 10 words to non-verbal) and motor skills (normal development to stereotypic movements) at 2 yo. At the time of the rs-fMRI, he was diagnosed with ASD, global developmental delay, severe LI, and DCD. Despite these deficits, FP, motor, and bilateral language RSNs were intact. Aberrant networks were localized to the FP, insular, and mesial temporal regions.

### 3.2. Patient Summaries of TLE Patients (Controls)

#### 3.2.1. Patient 6

A 15 yo female was diagnosed with intractable localization-related symptomatic epilepsy without status epilepticus after unprovoked complex partial seizures at 10 yo. The seizure focus was associated with an inferior temporal lobe encephalocele. Medical and developmental history was otherwise normal. The expected whole-brain network profiles were well-detected, including the motor, language, and FP networks. The SOZ was detected in the right temporal and right frontal regions. Non-SOZ aberrant networks were detected in the hand-arm motor areas.

#### 3.2.2. Patient 7 

An 11 yo female was diagnosed with intractable epilepsy without status epilepticus after complex partial seizures at 10 yo. The seizure focus was associated with a left temporal lobe tumor. Early medical history was notable for appendectomy. Early developmental language milestones were delayed. Expected whole-brain network profiles were well-detected, including the motor, language, and FP networks. The SOZ was detected in the anterior half of the anterior temporal lobe. Non-SOZ aberrant network features were detected in the facial motor and language areas.

#### 3.2.3. Patient 8

A 4 yo male had a history of complex febrile seizures starting at 20 months of age. The patient presented with typical motor and cognitive developmental milestones but had a diagnosis of expressive language disorder with poor articulation. Overall, the expected whole-brain network profiles were well-detected, including the motor, language, and FP networks. The SOZ was detected in the left mesial-anterior temporal regions (language). No non-SOZ aberrant networks were detected.

#### 3.2.4. Patient 9

A 9 yo male was diagnosed with intractable absence epilepsy without status epilepticus at 7 yo. Medical and developmental history was normal. The expected whole-brain network profiles were well-detected, including the motor, language, and FP networks. The SOZ was detected in the left mesial-temporal region. No non-SOZ aberrant networks were detected.

#### 3.2.5. Patient 10

A 13 yo male was diagnosed with intractable localization-related idiopathic epilepsy without status epilepticus at 12 yo. Medical and developmental history was normal. The expected whole-brain network profiles were well-detected, including the motor, language, and FP networks. The SOZ was detected in the right and left mesial-temporal regions. No non-SOZ aberrant networks were detected.

### 3.3. Statistical Analysis

Table 1 presents the clinical characteristics and rs-fMRI ICA-based network characteristics and the corresponding Fisher’s exact tests’ statistics.

#### 3.3.1. Clinical Characteristics

Children with ASD were between 4 and 16 yo (*M =* 10.66, *SD =* 4.71) and predominantly male (60%). Similarly, children with TLE were between 4 and 15 yo (*M =* 10.98, *SD =* 4.57) and predominantly male (60%). There were no significant differences in age (*t*(8) = 3.4, *p = 0*.916) or sex (Fisher’s exact test, *p =* 1.0) between the children with ASD and TLE. 

All children with ASD presented with clinically significant motor (100%), language (100%), and cognitive (100%) dysfunction, while children with TLE only presented with language dysfunction (40%) in addition to the epilepsy-related symptoms. Fisher’s exact test (two-tailed) indicated a significant relationship between motor and cognitive dysfunction and ASD diagnosis (Table 1).

#### 3.3.2. rs-fMRI Networks

ICA of rs-fMRI data yielded 91, 119, 49, 48, 58, 66, 105, 96, 95, and 111 ICs for patients 1–10, respectively. There was no significant difference (*t* (8) = 2.3, *p =* 0.212) in the mean number of ICs generated from subject-level ICA of rs-fMRI for the children with ASD (*M =* 73, *SD =* 31.09) and children with TLE (*M =* 94.6, *SD =* 17.3).

There was no significant association between typical networks and diagnosis, as all children with ASD and TLE presented with intact motor, language, and FP RSNs (Table 1). Children with ASD presented with non-SOZ aberrant motor (40%), language (80%), and FP (100%) networks, while children with TLE only presented with non-SOZ aberrant motor (40%) and language (20%) networks in addition to the identified SOZs. Thus, children with ASD were significantly more likely to manifest aberrant FP networks (Table 1).

#### 3.3.3. Aberrant rs-fMRI Networks and Clinical Symptomatology

The relationship between network characteristics and clinical dysfunction was assessed using the full study cohort (*n* = 10). There was a significant relationship between atypical (non-SOZ) FP networks and cognitive dysfunction (Table 2). Specifically, clinically significant cognitive dysfunction was documented in 100% of the children with detected atypical FP networks, while cognitive dysfunction was not reported (0%) in children without atypical (non-SOZ) FP networks.

## 4. Discussion

For the first time, we demonstrate that children with regressive-type ASD have intact motor, language, and FP neural networks with relatively typical spatial and temporal features, despite having moderate to severe disability in the skills typically subserved by these networks. The finding of overall preserved connectivity is consistent with prior ASD research [4,23]. Interestingly, the case sample only included children with regressive-type ASD, suggesting that these intact typical cognitive networks did subserve their normal expression prior to the regression. Furthermore, the rs-fMRI ICA data-driven approach extracts typical and atypical neural circuitry on an individual basis, demonstrating that multiple widespread aberrant neural networks characterize regressive-type ASD. Given that typical RSNs are intact but not well-expressed, we think that the atypical aberrant networks disrupt the fidelity of signaling within these long-range typical RSNs, essentially creating a locked-in network syndrome.

The ASD participants were found to have typical RSNs and additional broad aberrant networks. In comparison, the TLE controls had fewer aberrant networks beyond those localized to regions disrupted by seizure activity (SOZ). The aberrant networks found in ASD patients were not orthogonal to any canonical network, and this finding, in addition to their spatial location, may provide hints to the pathophysiology of ASD symptoms. In these patients, aberrant networks traversing regions associated with the FP network could interfere with the fidelity of signals of typical RSNs as they are communicated between distal regions. Higher-order networks typically integrate into other brain networks at around 18–24 months [24,25], which is around the same time children with ASD begin to show core symptoms and regression may occur [26]. According to our theory, interference from aberrant networks could impair the acquisition of cognitive inhibitory control [27], social skills, and other complex behavior. We think that interference from aberrant networks hinders communication and the optimal functioning of long-range typical RSNs, essentially creating a locked-in network effect. Alternatively, or complementarily, aberrant networks could inappropriately activate cortical areas. For example, aberrant networks traversing the somatosensory area, as in Patient 1, may be a biomarker of the expressed sensory symptoms.

Interventions that inhibit atypical networks could effectively “unlock” the intact RSNs, leading to symptomatic recovery. Recovery from locked-in syndrome, with compromised capacity to demonstrate consciousness yet intact supratentorial network function on rs-fMRI, was reported [28]. The patients presented in this study were all refractory to standard treatments, such as behavioral and speech therapy. Potentially, other network-targeted treatments, including those used in epilepsy, such as surgical and neuromodulatory treatments, could improve recovery rates of these patients [9,28,29]. 

This study has several limitations. The small sample size limits the generalizability of our results. It is also possible that the aberrant networks seen on imaging are an epiphenomenon and do not affect ASD symptoms. However, a recent large study found differences in the FP network regions between individuals with ASD and typically developing controls [4], thus supporting the notion that these aberrant networks interfere with typical RSNs. It is also possible that the aberrant networks reflect activity that does not interact with RSNs in an awake brain as our patients were sedated during scanning. However, such findings are not reported in neurotypical individuals studied under low-dose conscious sedation [30,31,32]. Further studies will be needed to correlate the presence or absence of aberrant networks and their characteristics, such as location, with detailed measures of cognitive and language function, as well as ASD symptomology, in larger cohorts. Unfortunately, with the current sample size, such analysis would not be valid. Thus, we look forward to larger studies in the future.

## 5. Conclusions

From these data, we propose that analyzing individual patients may provide evidence that ASD symptoms may correlate with aberrant networks that interfere with the maturation of typical RSNs, effectively creating a locked-in network syndrome. Further, we propose that it may be clinically useful to perform rs-fMRI on select patients with ASD. We believe this case series warrants larger systematic studies of ASD patients with rs-fMRI before and after typical treatment. rs-fMRI could help personalize treatment strategies by categorizing a patient’s aberrant networks based on treatment response and symptom profile. In this way, rs-fMRI can act as an integrative tool to support a personalized precision medicine approach to ASD diagnosis and treatment.

## 6. Patents

Nothing to report.

## Figures and Tables

**Figure 1 jpm-11-00854-f001:**
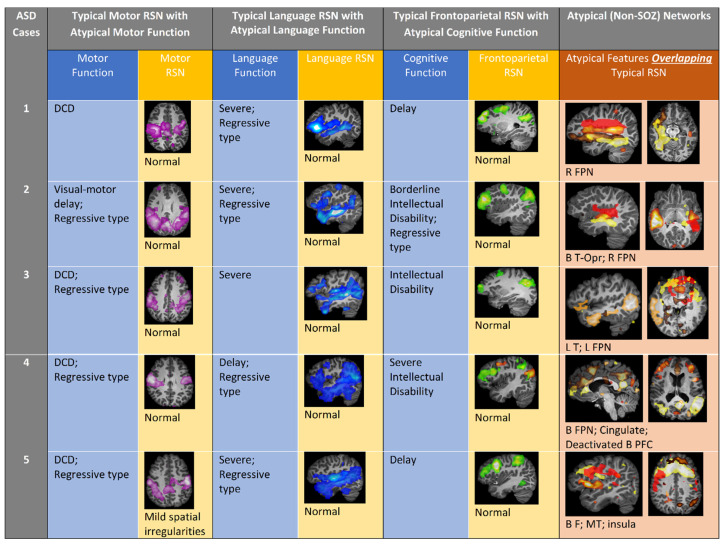
Comparison of clinical and rs-fMRI findings in ASD patients. Columns 1–3: ASD patient typical motor, language, and frontoparietal network images and interpretation with corresponding discordant phenotypic clinical impairments. Column 4: ASD participant atypical (aberrant) network (non-SOZ, overlapping typical RSN) images. B, bilateral; DCD, developmental coordination disorder; F, frontal; FPN, frontoparietal network; L, left; MT, mesial temporal; Opr, operculum; R, right; RSN, resting state network; vmPFC, ventromedial prefrontal cortex.

**Figure 2 jpm-11-00854-f002:**
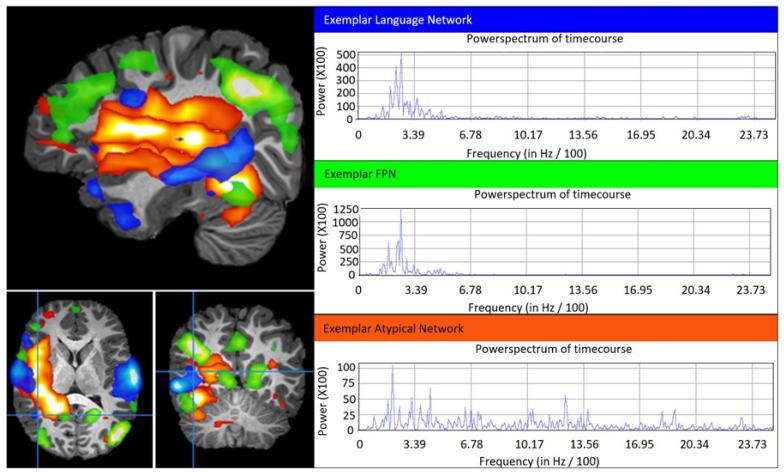
Example of right-sided language, frontoparietal, and aberrant networks in ASD Patient 1. Image shown is the sagittal, axial, and coronal T1-weighted MRI of Patient 1 with overlayed language (blue), frontoparietal (green), and aberrant (red) networks. Plotted are the respective network blood oxygen dependent signal (BOLD) power versus frequency (Hz/100), wherein typical is less than 6 Hz/100. The frontoparietal and language network spatial distribution and the BOLD power spectrum are typical, whereas the atypical networks have abnormal spatial and widely distributed power spectra.

**Table 1 jpm-11-00854-t001:** Clinical and resting-state network characteristics. Fisher’s exact test (two-tailed) was used to test the difference in frequencies. Significant *p* values (*p* < 0.017) are indicated in bold and italics.

**Characteristic/Network**	**Participants with ASD** **(*n* = 5)**	**Participants with TLE** **(*n* = 5)**	**Fisher Exact Test** ***p*-Value**
**Clinical Characteristics**			
Motor Dysfunction	100% (5)	0% (0)	** *0.008* **
Language Dysfunction	100% (5)	40% (2)	0.167
Cognitive Dysfunction	100% (5)	0% (0)	** *0.008* **
**Typical Resting-State Networks**			
Motor Network	100% (5)	100% (5)	1.000
Language Network	100% (5)	100% (5)	1.000
Frontoparietal Network	100% (5)	100% (5)	1.000
**Aberrant Resting-State Networks**			
Motor Network	40% (2)	40% (2)	1.000
Language Network	80% (4)	20% (1)	0.167
Frontoparietal Network	100% (5)	0% (0)	** *0.008* **

**Table 2 jpm-11-00854-t002:** Relationship between clinical symptomatology and presence of aberrant resting-state network. Fisher’s exact test (two-tailed) was used to test the difference in frequencies. Significant *p* values (*p* < 0.017) are indicated in bold and italics.

Aberrant Network	Clinical Dysfunction Subserved by Network	No Clinical Dysfunction Subserved by Network	Fisher’s Exact Test *p*-Value
Motor Dysfunction	100% (5)	0% (0)	1.000
Language Dysfunction	100% (5)	40% (2)	0.167
Frontoparietal Network	100% (5)	0% (0)	** *0.008* **

## Data Availability

Data are available upon request.

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
