# Peer review of "Locked-in Intact Functional Networks in Children with Autism Spectrum Disorder: A Case-Control Study"

_jpm, 2021, doi:10.3390/jpm11090854_

Round 1
Reviewer 1 Report
General Comments:
Authors compared 5 severe ASD patients with 5 TLE patients to identify whether brain network abnormalities are unique to ASD. They concluded that individuals with ASD appeared to have intact cognitive networks whose functions cannot be fully expressed potentially because atypical networks interfered with their long-range signaling, thus creating a unique “locked-in network” syndrome. Unfortunately, this manuscript has several major drawbacks including no healthy controls, very small sample sizes, questionable study design, missing critical information of methodology, lack of quantitative analysis, poor manuscript organization, and over-claimed conclusions. This study was not conducted with an appropriate design and the conclusions were mostly derived based on authors’ speculations rather than scientific evidence.
Specific Comments:
- While authors concluded that “ASD appeared to have intact cognitive networks…”, this may not be true without solid evidence provided in this study. Since 2012, there are over 500 published articles reporting altered or abnormal resting-state networks in ASD. Moreover, this study didn’t compare the ASD networks with healthy controls to confirm the abnormality. Instead of visual inspection on each single patient’s maps, what are the quantitative and statistical evidences for authors to conclude ASD patients have intact cognitive networks?
- In Figure 1, what are the criteria for authors to determine that most of the identified motor, language, and frontoparietal networks are normal in ASD patients? Similarly, what’s the definition or source of normal resting-state networks that overlapped on maps in the last two columns? Using results from TLE patients as normal RSNs is inappropriate.
- The presentation of Figure 1 should be adjusted. The contents of first three columns, specifically the clinical impairment degree, are lack of organization. Which type of impairment, to what degree of severity do authors want to address? The overlapping maps in the last two columns are complicated without proper explanation to the color labels and components.
- When deriving the conclusion like “ASD appeared to have intact cognitive networks whose functions cannot be fully expressed”, authors should provide a correlation analysis (using statistical approaches) to unravel the quantitative changes of RSN in ASD with patient’s symptoms or behavioral assessments. Without this scientific evidence, the conclusion is only a speculation.
- For the methodological concerns, critical information to understand and repeat this study was not provided. This should be carefully addressed and revised.
- What are the preprocessing steps applied to the BOLD fMRI? What are the utilized parameters?
- Which template for the spatial normalization was used? Was a pediatric brain template applied?
- Demographic data or clinical characteristics of TLE patients should also be provided.
- Which ICA algorithm (not software) was applied? What is the exact number of components for each patient? Was ICA performed in the group basis or subject basis? How authors identify the specific network from ICA results, by pure visual inspection or spatial similarity tests?
- Basically, there are obvious variations of RSN, even for the motor network, between healthy subjects reported in many studies. It is not allowed or appropriated to claim a normality of network in single subject. Most of the RSN patterns were identified based on a group study, i.e. the average of combination results of a certain number of subjects. Authors should carefully address this issue.
- Another major drawback of this manuscript is insufficient literature review in both Introduction and Discussion. Lack of the understanding in recent development in studying RSN in ASD may bias the study design and conclusion.
- Finally, I’d like to emphasize that a scientific report should base on the quantitative and statistical analysis to derive the conclusion. Over-interpretations should be carefully avoided.
Author Response
Specific Comments:
Q1. While authors concluded that “ASD appeared to have intact cognitive networks…”, this may not be true without solid evidence provided in this study. Since 2012, there are over 500 published articles reporting altered or abnormal resting-state networks in ASD. Moreover, this study didn’t compare the ASD networks with healthy controls to confirm the abnormality. Instead of visual inspection on each single patient’s maps, what are the quantitative and statistical evidences for authors to conclude ASD patients have intact cognitive networks?
R1. We agree that a growing body of research reports evidence of altered and abnormal resting state networks in ASD, often relative to a healthy control population. However, we take a different unique approach to examining resting state data. Rather than comparison resting state network as if they exist exactly the sample in all subjects, we recognize that there is normal variation in brain connectivity such that resting state networks may vary slightly from subject to subject. Thus, we take a subject (individual)level approach to examining resting state networks by extracting resting state networks from each individual’s brain. This allows us to examine the characteristics of typical resting state networks AND identify resting state network which are not typically found (aberrant networks). Our case-control study contributes to this body of research, presenting evidence of both typical and aberrant intact resting state networks at the subject-level in a clinical population. To address the reviewer’s comments related to the prior research, we have expanded our introduction to include additional references to relevant research findings (Line 54-81).
In this small-n clinical sample we opted to use a pathological control to investigate the association of network characteristics (typical and aberrant) with diagnosis (ASD and TLE). We did this for several reasons: 1) we have extensive expertise in subject-level independent component analysis (ICA) of rs-fMRI for the clinical evaluation and treatment planning of patients with TLE, and atypical network activity is well characterized in this population, 2) from a personalized-medicine perspective, we wanted to determine if network characteristics were unique to ASD, especially given the increased comorbidity of ASD and TLE, 3) this is a retrospective study that utilized standard of care data within a clinical population. However, we agree that comparing ASD to healthy controls is the best way to determine pathological versus normal findings. In this regard, to compare against “norms”, we used expert classification of independent components (ICs) according to norms of resting state network spatial and temporal criteria, which is the gold standard for automated ICA of resting state fMRI. This is clarified in (Line 170-172), “IC categorization followed the working paradigm previously published [10], separating patient ICs into four categories – noise, typical RSNs, SOZ, and atypical networks – using criteria modified from established norms [10,14-17,23].” We agree that the rationale for choosing a pathological control group (pediatric epilepsy) for comparing ASD should be stated more clearly. We now emphasize that the utility of clinical application of any new biomarker must take the step beyond simple comparison to a heathy control group, to distinguishing against other pathologies. For example, if the signature of epilepsy is the same as ASD, then such biomarkers may lead toward misguided therapies. Thus, we have now modified Methods, paragraph 1, sentence 2 (line 122-124) to state, “a pathological control group was selected to distinguish ASD-specific atypical rs-fMRI biomarkers from known TLE-specific markers, given the increased comorbidity of these conditions.”
To address the reviewer’s comments on the visual inspection approach and lack of statistical evidence to support our conclusions, we have significantly revised our methods section to include a description of the criteria and prior works used to the develop the independent component (IC) categorization via visual inspection workflow (Line 109-142; line 153-204). This includes both quantitative and qualitative criteria for IC determinations. Further, this is not a group-level ICA design. Rather, the ICs are extracted and classified at the subject-level and the clinical findings are transformed into categorical variables for statistical analysis with the Fisher’s exact test (line 206-213).
We recognize that this case-control design and resultant analysis and conclusions have limited level of evidence, as stated in the last paragraph of the discussion (Line 381), sentence 1, “This paper has several limitations. The small sample size limits the generalizability of our results.” To emphasize this and the editor’s constructive points we have now also modified the abstract to highlight the subject-level analysis (Line 31), statistical evidence (Line 39-43), and modified the conclusion statement (Line 43-49) to now read, “Despite severe cognitive delays, children with regressive type ASD may demonstrate subject-level intact typical cortical network activation despite an inability to use these cognitive facilities. The functions of these intact cognitive networks may not be fully expressed, potentially because aberrant networks interfere with their long-range signaling, thus creating a unique “locked-in network” syndrome.”
Q2. In Figure 1, what are the criteria for authors to determine that most of the identified motor, language, and frontoparietal networks are normal in ASD patients? Similarly, what’s the definition or source of normal resting-state networks that overlapped on maps in the last two columns?
R2. We agree that the normative resting state network comparison benefits from further clarification, and this question is in line with Q1, and clarification is now made as detailed in the R1. response above.
Q3. Using results from TLE patients as normal RSNs is inappropriate.
R3. As above, we agree the study design reasoning to TLE would benefit from clarification, as distinction of biomarker results between common pathologies is needed for practical clinical application and now addressed in the revised manuscript as detailed in R1.
Q4. The presentation of Figure 1 should be adjusted. The contents of first three columns, specifically the clinical impairment degree, are lack of organization. Which type of impairment, to what degree of severity do authors want to address? The overlapping maps in the last two columns are complicated without proper explanation to the color labels and components.
R4. We agree that Figure 1 presents a great deal of information, and achieving optimal clarity is not trivial. We have modified the figure to make the column headings clearer (see below). We have also removed the column 5, as it presented SOZ-related atypical networks in the TLE group (not a primary aim of the study). Finally, we expanded the patient descriptions of the case and control subjects to provide context to this summary figure.
Column 1 – Typical Motor RSN with Atypical Motor Function
Column 2 – Typical Language RSN with Atypical Language Function
Column 3 -Typical Frontoparietal RSN with Atypical Cognitive Function
Column 4 – Atypical (Non-SOZ) Networks
The reviewer commented that the “type of impairment” and “degree of severity” is not optimal. The wording we currently have is primarily around “regressive” and “severe”, though we expand on the individual differences in patient summaries. Looking further, each subject has at least one stream of development affected in the severe or regressive category. The presence or absence of clinical impairment was transformed into a categorical variable and compared between both groups. This investigation did not specifically investigate the association between network characteristics and clinical function. Rather, we highlight the juxtaposition of the patient clinical summaries and rs-fMRI network findings of intact typical networks.
Q5. When deriving the conclusion like “ASD appeared to have intact cognitive networks whose functions cannot be fully expressed”, authors should provide a correlation analysis (using statistical approaches) to unravel the quantitative changes of RSN in ASD with patient’s symptoms or behavioral assessments. Without this scientific evidence, the conclusion is only a speculation.
R5. We agree that the study rigor improved by statistically examining the association between the behavior and resting state networks on an individual level.
Q6. For the methodological concerns, critical information to understand and repeat this study was not provided. This should be carefully addressed and revised.
What are the preprocessing steps applied to the BOLD fMRI? What are the utilized parameters?
Which template for the spatial normalization was used? Was a pediatric brain template applied?
R6. We agree that detailed description of all methodological parameters is the best way forward for optimal reproducibility. Therefor our sequence, preprocessing, utilized parameters, thresholding is now detailed. Regarding spatial normalization and utilization of brain template, because our study design was intended to forward the science of precision medicine, group averages and molding of patient brains to a template was not performed. Rather all analysis was performed in subject space, which is now clarified in the methods.
Q7. Demographic data or clinical characteristics of TLE patients should also be provided.
R7. We agree that the TLE patient controls demographics and clinical characteristics should be made clear. This information is now included in separate patient summaries and in Table 1.
Q8. Methods.
Q8a. Which ICA algorithm (not software) was applied?
R8a. This information and appropriate citations were included in the methods section.
Q8b. What is the exact number of components for each patient?
R8b. This information was included in the patient summaries and the group averages included in Table 1.
Q8c. Was ICA performed in the group basis or subject basis?
R8c. ICA was performed on a subject-level basis. This was clarified in the methods.
Q8d. How authors identify the specific network from ICA results, by pure visual inspection or spatial similarity tests?
R8d. We used a documented/published visual inspection categorization procedure based on normative spatial values. Relevant publications are cited, procedures stated, and a description of the categorization criteria are stated in the methods section.
Q9. Basically, there are obvious variations of RSN, even for the motor network, between healthy subjects reported in many studies. It is not allowed or appropriated to claim a normality of network in single subject. Most of the RSN patterns were identified based on a group study, i.e. the average of combination results of a certain number of subjects. Authors should carefully address this issue.
R9. We agree that most rs-fMRI analysis are performed on the group level and result in ranged of connectivity values, thereby providing normative data for the given population. We further agree that the term “normal” was not clearly defined. Therefore, we have modified our language to use the term “typical” rather than “normal” and we operationally define “typical” in regards to individual resting state networks to the Methods section, paragraph 5, last sentence, “Rs-fMRI data was interpreted by a rs-fMRI specialist (senior author) wherein typical/normal RSNs are operationally defined as those meeting spatial and temporal criteria above by expert visual inspection.”
Q10. Another major drawback of this manuscript is insufficient literature review in both Introduction and Discussion. Lack of the understanding in recent development in studying RSN in ASD may bias the study design and conclusion.
R10. We agree, to address these your comments related to the prior research we have expanded our introduction to include additional references to relevant research findings including a large meta-analysis and large database validation study. (Line 54-81).
Q11. Finally, I’d like to emphasize that a scientific report should base on the quantitative and statistical analysis to derive the conclusion. Over-interpretations should be carefully avoided.
R11. We agree that increasing quantitative and statistical rigor would improve our methods and strengthen the reproducibility. Thus, we have addressed this in the prior questions and responses above. To summarize, we expanded the description of the ICA and statistical analysis procedures. We included additional patient summaries and a table of all group means/counts and statistical outcomes.
Reviewer 2 Report
Autism Spectrum Disorder (ASD), which is also referred as "developmental disorder", is associated with individual’s communication and behaviour problems, presently attract lots of attention from research and medical personnel, especially during the last decade. ASD affects a high percentage of worldwide population, and the true neurophysiological mechanism involved into on-set and progression of this neurodegenerative disorder, are not well understood. The reviewed manuscript represents a valuable data received using the Magnetic Resonance Imaging (rs-fMRI), which analysed the potential abnormalities in brain network structure and connectivity in individuals by ASD. It has been found that the atypical neuronal networks in ASD patients creates “locked-in network” circuits, which lead to luck of expressing and normal outcome, usually generated by the intact cognitive networks. In spite of a necessity to have a larger study involving a much larger number of individuals affected by ASD, the manuscript represents a valuable contribution to a deeper understanding of how neuronal networks are functioning in patients with this particular medical condition, and data received are pointing towards an opportunity to develop a potential individualised intervention, which will be able to supress atypical neuronal circuits, and could possibly lead to a symptomatic recovery.
Author Response
Response. Thank you for your feedback. We agree that our small-n clinical sample is a limitation of the current study and reduces the generalizability of the findings. Based on this pilot investigation we hope to conduct a large-scale study using our clinical patient population (children with/without regressive type ASD) and publicly available data (ASD and healthy controls).
We hope that the requested revisions add clarity to our methodology, provide evidence to support and strengthen our conclusions, and further the development and monitoring of individualized interventions.
Round 2
Reviewer 1 Report
I thank the authors for providing a thoroughly revised version of the manuscript. They have addressed my previous comments carefully and improved the quality of manuscript. With the sufficient statistical evidence, the findings become more convincing now.
However, I highly suggest authors to avoid a large number of self-citations, specifically the Ref. 8-12. I don't see the necessity for citing all of these previous works of authors. It raises my concern of inappropriate self-citations. Please do remove them before my recommendation for the publication.
Author Response
We apologize for any perceived self-citations, we were simple trying to emphasize our experience with this technique. We have removed 2 of the 5 references from this sentence.